# Morphological and Optical Coherence Tomography Aspects of Non-Carious Cervical Lesions

**DOI:** 10.3390/jpm13050772

**Published:** 2023-04-29

**Authors:** Andreea Stănuşi, Monica Mihaela Iacov-Crăițoiu, Monica Scrieciu, Ioana Mitruț, Bogdan Costin Firulescu, Mihaela Roxana Boțilă, Diana Elena Vlăduțu, Adrian Ştefan Stănuşi, Veronica Mercuț, Eugen Osiac

**Affiliations:** 1Department of Prosthetic Dentistry, University of Medicine and Pharmacy of Craiova, 200349 Craiova, Romania; andreea.stanusi@umfcv.ro (A.S.); monica.scrieciu@umfcv.ro (M.S.); ioanamitrut@gmail.com (I.M.); 2University of Medicine and Pharmacy of Craiova, 200349 Craiova, Romania; dr.bogdanfirulescu@gmail.com (B.C.F.); mihaelabotila09@yahoo.com (M.R.B.); dianavladutu04@gmail.com (D.E.V.); adrian.stefan.stanusi@gmail.com (A.Ş.S.); 3Department of Biophysics, University of Medicine and Pharmacy of Craiova, 200349 Craiova, Romania; eugen.osiac@umfcv.ro

**Keywords:** non-carious cervical lesions, wedge shaped, optical coherence tomography

## Abstract

Non-carious cervical lesions (NCCLs) are considered the irreversible losses of dental hard tissues at the cemento–enamel junction, in the absence of acute trauma and dental caries. The aim of this study was to highlight the presence of NCCLs in cervical areas based on specific macroscopic aspects in order to establish their clinical form, size and location and to confirm the role of optical coherence tomography (OCT) examination in the early diagnosis of these lesions. For this study, 52 extracted teeth were used, which did not have endodontic treatments, fillings or carious lesions in the cervical area. All teeth were examined macroscopically and OCT was used to evaluate the degree of occlusal wear, the presence and clinical form of NCCLs. Most NCCLs were identified on the buccal surfaces of the premolars. The most frequently encountered clinical form was the wedge-shaped form, with a radicular location. NCCLs present most frequently in the wedge-shaped form. Teeth that presented several NCCLs were identified. The OCT examination is an adjunct method to evaluate the clinical forms of NCCL.

## 1. Introduction

Non-carious cervical lesions (NCCLs) are defined as the irreversible loss of dental hard tissues at the cemento–enamel junction, in the absence of acute trauma and the infectious component attributed to carious pathologies [1,2]. Currently, these lesions are considered to have a multifactorial etiology, which includes biocorrosion (erosion), friction (abrasion) and stress (abfraction) [1,3,4].

The non-carious loss of hard dental tissues was first mentioned in the literature by Dr. Hunter in 1778 [5]. In 1894, Zsigmondy described the wedged-shaped lesions located in the cervical area of the teeth, preferentially on the buccal surfaces of anterior maxillary teeth [6]. In 1991, Grippo [7] introduced for the first time the term “abfraction” to name these lesions.

In 2019, a group of experts from the European Organization for Research on Dental Caries and the Cariology Research Group of the International Association for Dental Research (IADR) met in a workshop organized in Frankfurt with the aim of defining the terminology of erosive tooth wear and dental caries. They unanimously established that the term “abfraction” should not be used anymore because there is not enough information to demonstrate that this form of dental wear is determined by a single process [8]. Despite their recommendation, the term is still used, suggesting the role of occlusal stress in the etiology of these lesions [9,10,11,12].

NCCLs can be present in all age groups, but clinical studies have shown an increase in the prevalence and severity of these defects in elderly patients, by summing up the effects of etiological factors [13,14,15]. The prevalence of NCCL has an average worldwide value among adult patients of 46.7% and varies between 5 and 85% [2]. The large variation in the prevalence reported in studies from specialized literature can be attributed to the examination techniques used and the criteria for classifying the lesions. Some studies refer to NCCLs as a whole, while other studies separately report cervical erosions, cervical abrasions and abfractions. According to Rees [16] and Grippo [17], all these clinical forms should be included in NCCLs.

Thus, for the prevalence of NCCLs as a whole, the studies carried out by Zi Yun Lai in 2015 [18], Que in 2013 [19], Hirata in 2010 [20] and Jakupovic in 2010 [21], are relevant in terms of the number of evaluated subjects and age groups.

NCCLs particularly influence the patient’s oral status, by affecting the dental aesthetics, producing dentine hypersensitivity, and because of the evolution of these lesions, the risk of teeth fracture increases [22,23,24].

Although several strategies have been proposed for the management of NCCLs, in practice they are applied differently, most likely due to the lack of knowledge of the etio-pathogenesis and prognosis of these lesions with/without treatment [25,26].

The diagnosis of NCCLs is made based on an examination by visual inspection and various magnification systems (loupe, microscope, intra-oral camera) and are of real use [27,28,29]. Grippo and Soares proposed the description of NCCLs based on the characteristics of macro- and micromorphology [30].

Macromorphology includes aspects that can be identified clinically, through direct visualization of the NCCL, such as shape (geometry), size and location. Regarding the location of NCCLs, most specialists have described the possibility of finding these lesions especially on the buccal surfaces. Few studies have identified NCCLs on other axial surfaces, or several such lesions on the same tooth [7,25,31,32].

Micromorphology includes aspects that can be identified through histological and microscopic examinations of the tissues affected by these lesions, such as the surface texture and the presence/absence of sclerotic dentin [30].

The optical coherence tomography (OCT) investigation is a new, adjunctive, non-invasive imaging method, which has attracted the interest of specialists due to the multiple benefits it brings to the examination of hard dental structures. Through the OCT examination of teeth, data can be obtained about the quantity and quality of dental hard tissues, in real time, without X-ray irradiation, and micromorphological elements of dental lesions can also be observed [33,34]. Through OCT examination, it is even possible to differentiate the various forms of dental wear [35]. OCT was introduced in dentistry for the study of hard and soft tissues. Currently, it can be used in in vivo or in vitro studies, to highlight quantitative and qualitative changes in the tissues of the oral cavity, having applications in the early diagnosis of dental caries, dental wear and the progression of periodontal disease [36]. The OCT examination can even be used in the early diagnosis of oral cancer [35].

However, there are few studies that use OCT examinations for the diagnosis of NCCLs, especially by examining the cervical area on all axial surfaces, due to the limited accessibility of this medical device [31,37,38,39].

The aim of this study was to highlight the presence of NCCLs in cervical areas based on specific macroscopic aspects in order to establish their clinical form, size and location and to confirm the role of OCT examination in the early diagnosis of these lesions.

## 2. Materials and Methods

### 2.1. Teeth Prevelation and Preparation

The study was carried out on 112 teeth that were extracted from 56 patients aged between 30 and 65 years, who presented themselves for diagnosis and treatment at the Dental Prosthetics Clinic of the Faculty of Dentistry, University of Medicine and Pharmacy of Craiova, in January–February 2023. The extracted teeth were: 28 lower incisors, 12 upper incisors, 2 upper canines, 3 lower canines, 20 upper premolars, 16 lower premolars, 13 upper molars and 18 lower molars.

The study was approved by the Ethics Committee of the University of Medicine and Pharmacy from Craiova, Romania (no.55/16.02.2023). All patients included in the study provided written informed consent regarding the dental treatment presented and for the participation in the study.

The inclusion criteria were: (1) the patient’s consent to participate in the study, (2) the teeth to be extracted had severe damage to the periodontal tissues or various dental diseases for which conservative treatment could not be performed.

The exclusion criteria were: the presence of fillings, endodontic treatment or carious lesions in the cervical area.

For the study we atraumatically extracted teeth, and then disinfected them by keeping them in a peroxide solution 10% for 10 min to eliminate organic residues. Each tooth was cleaned by ultrasonic scaling and curettage. The teeth were kept in NaCl 0.9% until the examinations were carried out to avoid their dehydration [31,35]. Extractions, disinfection, cleaning and storing the teeth were carried out by 2 dentists, a coordinator and an operator, specialists in the field of dental wear, respecting the described work protocol for each tooth.

Each tooth was examined macroscopically, at the occlusal surface and in the cervical areas of the axial surfaces, and subsequently, was examined using OCT at the level of the cervical areas.

### 2.2. Macroscopic Examination

At the level of the occlusal surfaces, the presence of lesions and the wear score was recorded based on the Smith and Knight index [40].

At the level of the cervical surfaces, the presence of NCCLs was recorded. For the teeth that presented NCCLs, the affected axial surface, the anatomical locations of the lesions, their shape and depth were recorded.

The location of the NCCL was assessed based on the recommendations of Grippo and Soares, who referred to evaluating the degree of interest in the dental hard tissues and the location at the level of the crown and/or root [30].

To determine the shape of the NCCL, the walls of the lesion were identified: the coronal wall, from the occlusal surface of the tooth and the gingival wall, from the vicinity of the cervical region [30]. Depending on the angle formed by the two walls, the lesions were classified into two patterns: wedge shaped with a sharp angle and saucer shaped with a rounded angle [3,22,34].

The depth of the NCCL was assessed based on the perpendicular starting from an imaginary line which is considered to be parallel to the long axis of the tooth, which joins the coronal and gingival edges of the lesion, to the deepest point of the lesion [21,30]. The depth measurement was made by the same operator using a millimeter-graduated periodontal probe.

For each tooth, during the macroscopic examination, photos of the occlusal surface and axial surfaces were taken using a Canon DSLR 600EOS (Tokyo, Japan). The photos were stored as JPG images. All photos were taken, processed and interpreted by the same operator. 

### 2.3. OCT Examination

For the OCT investigation of the cervical areas of the selected teeth, each tooth was fixed in silicone material of high consistency (Zetaplus L Intro KIT, Zhermack, Badia Polesine, Italy), in such a manner that only one surface at a time was exposed for the OCT examination. To carry out this study, the Thorlabs Swept Laser Source OCT system (OCS1300SS, Thorlabs) was used, which has a laser beam with a spectral band of 100 nm, a wavelength of 1310 nm and an average power of 12 mW [41,42]. The examination of the samples was carried out on a depth of 2.5 mm and a length of 10 mm. The OCT examination was performed by a specialist in the field of biophysics.

For each tooth, during the OCT examination of the cervical areas, 2D OCT images were obtained, which were stored as JPG files. The images were interpreted by 3 operators: the biophysics specialist and 2 dental wear specialists.

The criteria for identifying NCCLs by OCT examination were lack of OCT signal at the periphery of dental hard tissues; areas of structural alteration of dental hard tissues in the form of a bright area, on a gray background.

The data obtained through the macroscopic and OCT investigation of the selected teeth were centralized and interpreted in the Microsoft Excel program.

## 3. Results

Considering the inclusion and exclusion criteria, it was found that 52 teeth presented NCCLs and cracks in the cervical area. Of these, 28 teeth were incisors (8 upper incisors and 20 lower incisors) and 24 teeth were premolars (14 upper premolars and 10 lower premolars).

### 3.1. Macroscopic Examination of Teeth with NCCLs

After the macroscopic examination, 21 teeth (10 incisors and 11 premolars) were found to have NCCLs. Figure 1 shows the most suggestive images for the clinical forms of NCCLs.

The morphological aspects recorded by macroscopic examination of the 21 teeth with NCCLs are presented in Table 1. 

Through macroscopic examination, 27 NCCLs were identified (5 teeth presented several NCCLs), of which 11 (40.74%) were on incisors and 16 (59.26%) on premolars. It was observed that 5 teeth presented several macroscopically visible NCCLs (Figure 2).

The most frequent location was on the buccal surface, where 18 NCCLs were identified (66.67%), followed by the oral face with 7 NCCLs (25.93%) and the proximal faces with 1 NCCL each (3.70%).

From an anatomical point of view, 13 NCCLs had a root location (48.15%), 11 NCCLs were involved in both crown and root tissues (40.74%) and 3 NCCLs had a crown location (11.11%).

Regarding the shape of the NCCLs, 18 of them were wedge shaped (66.67%) and 9 were saucer shaped (33.33%).

Regarding the depth of the NCCLs, 22 lesions had a depth of less than 1 mm (81.48%), 3 lesions had a depth greater than 2 mm (11.11%) and 2 lesions had a depth between 1 and 2 mm (7.41%).

### 3.2. OCT Examination of Teeth Included in the Study

All 52 teeth included in the study were examined (21 teeth with macroscopically visible NCCLs and 31 teeth without macroscopically visible NCCLs).

A maximum of 512 OCT images was taken for each dental surface examined. Figure 3, Figure 4 and Figure 5 present specific OCT images of teeth with macroscopically identified NCCLs, but also specific OCT images of teeth in which the NCCLs were not visible macroscopically. In each figure depicting the OCT images obtained, the position of the dental crown is marked with the letter C, the position of the root is marked with the letter R, and the arrow indicates the NCCL.

Centralizing the data obtained based on the OCT examination, seven more NCCLs were identified on six teeth (Table 2).

Through macroscopic and OCT examination, 34 NCCLs were identified, of which 14 were in incisors (41.18%) and 20 in premolars (58.82%). It was observed that, in the end, 6 teeth (1 incisor and 5 premolars) presented several NCCLs that were visible macroscopically and on OCT examination (Figure 6).

The premolars were the most affected teeth. They represented 46.15% of the total number of teeth selected for examination, and after the macroscopic and OCT examination, it was found that they represented 51.85% of the total number of teeth with NCCLs, followed by incisors which represented 53.85% when included in the study and with a proportion of impairment of 48.15%.

The most frequent location of NCCLs was on the buccal surface, where 21 NCCLs (61.76%) were identified macroscopically and by OCT, followed by the oral surface with 7 NCCLs (20.59%) and the proximal surfaces with 3 NCCLs each (8.82%) (Figure 7a). 

From an anatomical point of view, of all the lesions identified macroscopically and OCT, most NCCLs (19 NCCLs) were found on the root (55.88%), 12 NCCLs affected both the root and crown (35.3%) and 3 were found on the crown (8.82%) (Figure 7b).

Regarding the shape of the NCCLs identified macroscopically and by OCT, most NCCLs (20 NCCLs) were wedge shaped (58.82%), 13 were saucer shaped (38.24%) and 1 was irregular (2.94%) (Figure 8a).

Regarding the depth of the NCCLs identified macroscopically and by OCT, most NCCLs (27 NCCLs) had a depth of less than 1 mm (79.41%), 4 had a depth between 1 and 2 mm (11.77%) and 3 had a depth greater than 2 mm (8.82%) (Figure 8b). 

## 4. Discussion

NCCLs represent a dental condition with a complex etiology and treatment, which still arouses the interest of specialists, a fact demonstrated by the large number of publications and studies in specialized literature. From 2018 to now, more than 150 studies have been published in PubMed indexed journals regarding this form of dental wear. Most of the studies focused on the etiology of NCCLs [43,44,45,46], but there is currently no established consensus regarding this of for potential treatments, due to the frequent failures recorded in the practical studies [4,47,48,49].

The aim of this study was to identify the presence of NCCLs based on clinical aspects and to confirm the role of OCT in establishing the diagnosis of these lesions.

NCCLs represent a form of dental wear with a high prevalence among elderly patients. Although theoretically, any tooth can develop an NCCL, it has been observed that they are most frequently located on premolars [50,51,52]. The results of the presented study support this statement, with more than half of the NCCLs found on premolars. The preferential involvement of premolars by NCCLs is considered to be the result of the naturally imperfect morphology of these teeth and the high occlusal forces in the posterior region of the dental arches [51].

These lesions are most frequently located on the buccal surfaces of teeth, sometimes on the oral ones and rarely on the proximal ones [31,32,51]. In the presented study, most NCCLs were identified in buccal cervical areas, followed by oral and proximal ones. The preferred location of NCCLs on the buccal surfaces of teeth can be associated with the action of excessive occlusal forces with a para-axial direction [23].

The struggle that practitioners encounter in understanding the etiology and the optimal therapeutic protocol for NCCLs is supported by the diversity of their clinical forms [25,32,53].

The majority of specialists have stated that the predominance of one of the three main etiological mechanisms (friction, biocorrosion and stress) conditions the shape of the lesion [1,34]. The wedge shape of some NCCLs was considered indicative that abfraction/stress is the main etiological agent [24,54]. The saucer shape of some NCCLs is generally shallower and is considered to be determined by erosion [34]. The clinical form of the injury influences its progression. The saucer-shaped lesions progress by increasing in height, while the wedge-shaped lesions grow both in height and in depth [55].

NCCLs can have combined aspects of these two types [30]. Certain teeth can be affected by both types of lesions. Since these lesions can have two or more internal angles, some authors have defined them as “irregular” [56]. Over time, other terms have been used to describe the multitude of clinical forms of NCCL: superficial, concave, notched, irregular, dished out, cup shaped, letter shaped, superficial grooves, angular-irregular, curved irregular and angular- and curved-irregular [2,56].

In the presented study, wedge-shaped, saucer-shaped and irregular NCCLs were identified by macroscopic and OCT examination. Most NCCLs were wedge-shaped, followed by saucer-shaped ones. One lesion with an irregular shape was identified by OCT examination. The fact that the majority of NCCLs were wedge-shaped shows that the role of occlusal stress in the development of these lesions is very important and a therapeutic protocol must be applied to reduce its involvement in the occurrence/progression of the lesions.

It was observed that following the OCT examination of the dental cervical areas, seven NCCLs were identified that were not noticed during the initial examination of the teeth. These lesions were superficial and located mainly on the proximal surfaces of the teeth. The identification of these NCCLs by OCT examination draws attention to the importance and accuracy of this device in the diagnosis of dental defects. The OCT investigation allows the identification of small morphological changes, by obtaining enlarged 2D images of the examined tissues. Because of this function, incipient NCCLs could be identified, which were not detected by simple visual inspection during macroscopic examination. 

Regarding the involvement of dental hard tissues, NCCLs can affect only the dental crown, only the root, or they can extend into crown and root tissues. For the latter ones, the difficulties during adhesive restorative treatments are known, such as obtaining direct clinical access and the appropriate isolation of the operative field [57,58]. In the presented study, most lesions had a radicular evolution. The radicular evolution of the NCCLs identified in this study can be associated with the inferior mechanical properties of the radicular cement compared to dental enamel.

When describing NCCLs, certain parameters are also used, such as the angle formed between the walls of the cavity and the depth of the lesion. The depth of any carious or non-carious lesion is inversely proportional to the thickness of the remaining dentin, thus becoming an important factor in determining the pulp status [59]. In the presented study, most of the lesions had a depth of less than 1 mm. These superficial lesions can often go unnoticed by the dentist who does not use a magnifying system. The detection of small, superficial lesions is improved by the use of OCT, which allows both the identification of substance losses and their measurement.

The presented study draws attention to the diversity of clinical forms of NCCLs and to the fact that the early, superficial ones can be left unnoticed during clinical examination. 

The present research brings, in addition to previous research, the highlighting of several NCCLs in the same tooth. The authors recommend OCT examination for the accurate assessment of dental hard tissue defects.

The weak points of this study include the small number of teeth examined, but the inclusion in the study was limited by the indication for tooth extraction for periodontal reasons. Furthermore, after applying the inclusion and exclusion criteria, the study could not be performed on canines and molars, limiting the examination of NCCL aspects to premolars and incisors.

## 5. Conclusions

The study highlighted several clinical forms of NCCLs through macroscopic examination and OCT.

The OCT examination highlighted NCCLs that could not be identified by macroscopic examination, proving its effectiveness in the early diagnosis of these lesions.

The most frequent localization of NCCLs was on the buccal surfaces of the premolars, in the cervical area, with a radicular evolution, but NCCLs were also identified on other dental axial surfaces.

Teeth that presented several NCCLs were identified. The finding of NCCLs on other axial surfaces showed the distribution of stress in these areas, as a result of the occlusal loading on teeth from other directions, which predisposes them to fractures.

The deepest NCCLs were wedge shaped and located on the buccal surface with a radicular evolution, highlighting the role of occlusal overloads in the genesis of these lesions.

According to these conclusions, the management of NCCLs must consider, in addition to restoring the lost dental hard tissues, ensuring occlusal stability.

## Figures and Tables

**Figure 1 jpm-13-00772-f001:**
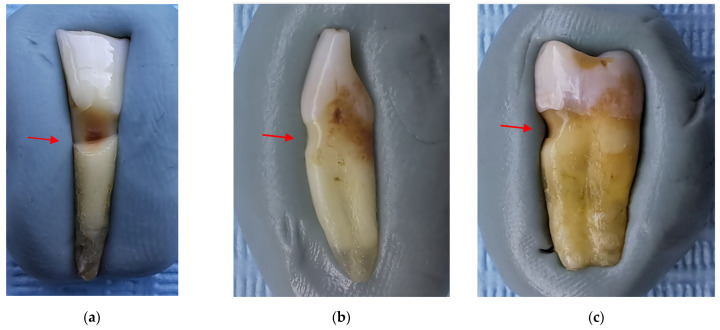
Macroscopic aspects of NCCL: (**a**) wedge-shaped NCCL indicated by the red arrow; (**b**) saucer-shaped NCCL indicated by the red arrow; (**c**) irregular NCCL indicated by the red arrow.

**Figure 2 jpm-13-00772-f002:**
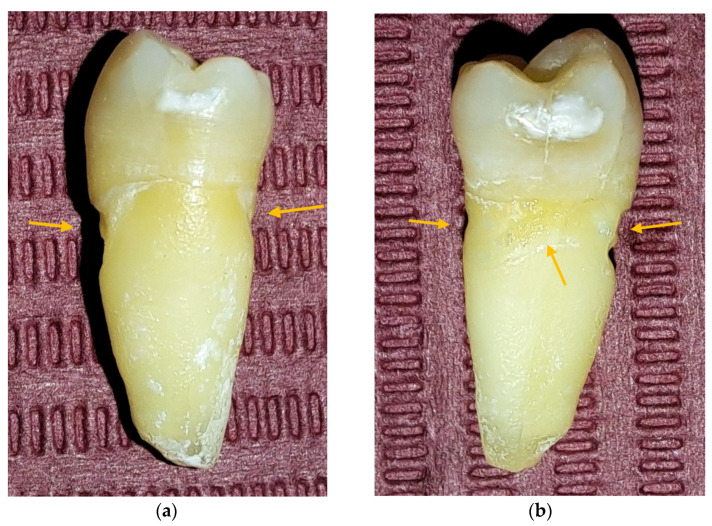
PM 15 macroscopic aspects: (**a**) macroscopic image of the mesial surface in which NCCLs are observed on the buccal and oral surface, indicated by the arrows; (**b**) macroscopic image of the distal surface in which NCCLs are observed on the buccal, oral, and distal surface, indicated by the arrows.

**Figure 3 jpm-13-00772-f003:**
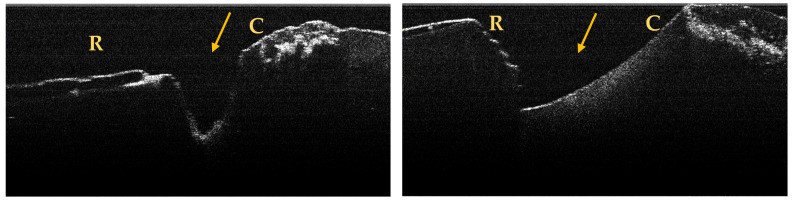
OCT images of a wedge-shaped NCC, indicated by the arrows.

**Figure 4 jpm-13-00772-f004:**
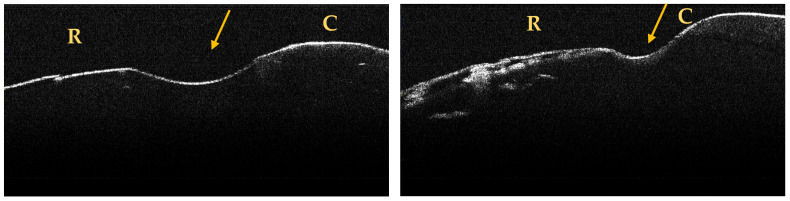
OCT images of a saucer-shaped NCCL, indicated by the arrows.

**Figure 5 jpm-13-00772-f005:**
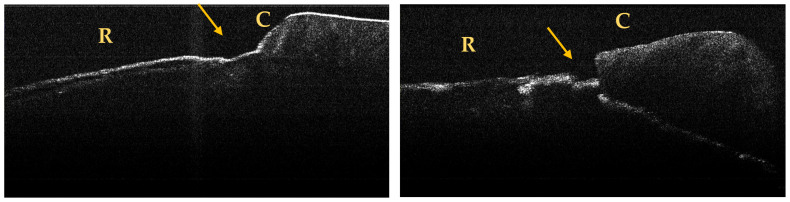
OCT images of an irregular NCCL, indicated by the arrows.

**Figure 6 jpm-13-00772-f006:**
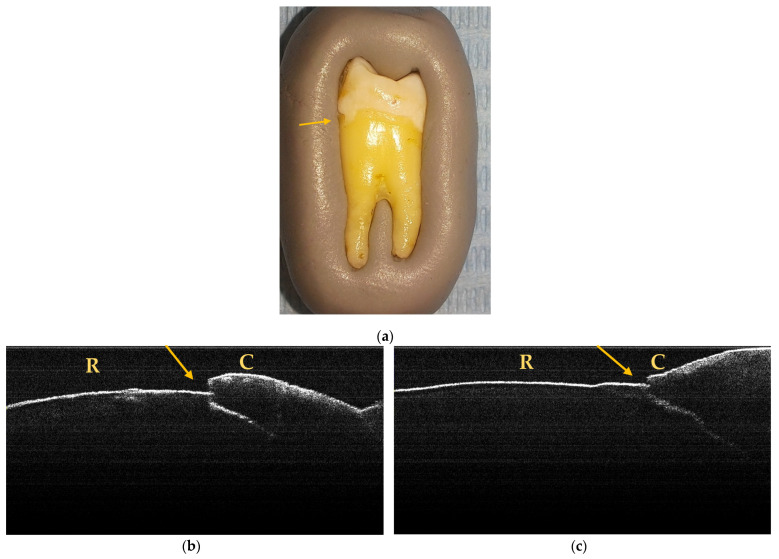
Macroscopic and OCT aspects of PM6: (**a**) macroscopic image showing superficial NCCL on the buccal surface, indicated by the arrow; (**b**) OCT image of the buccal surface where an NCCL is observed, indicated by the arrow; (**c**) OCT image of the mesial surface where an NCCL is observed, indicated by the arrow.

**Figure 7 jpm-13-00772-f007:**
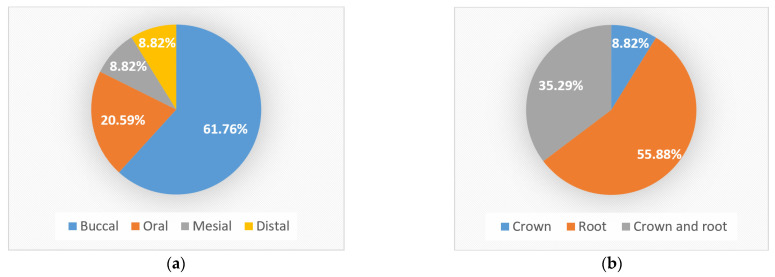
Distribution of NCCLs: (**a**) on surfaces; (**b**) anatomical location.

**Figure 8 jpm-13-00772-f008:**
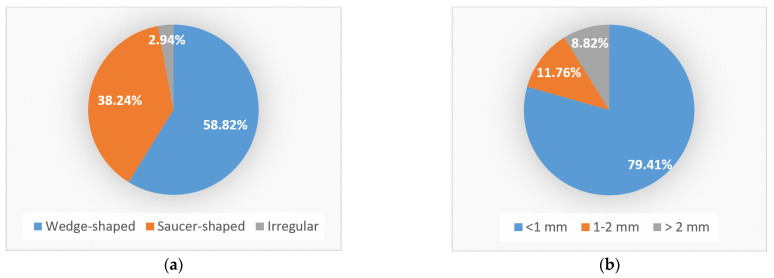
Distribution of NCCLs: (**a**) according to shape; (**b**) according to depth.

**Table 1 jpm-13-00772-t001:** NCCL aspects recorded through macroscopic examination.

No.	No. NCCL/Tooth	Tooth with NCCL	Surface with NCCL	Anatomical Location of NCCL	NCCL Shape	NCCL Depth
1	2	I6	Buccal	Crown and root	Wedge shaped	<1 mm
			Oral	Crown and root	Wedge shaped	<1 mm
2	1	I7	Buccal	Crown and root	Wedge shaped	>2 mm
3	1	I8	Buccal	Crown and root	Wedge shaped	<1 mm
4	1	I9	Buccal	Crown and root	Wedge shaped	>2 mm
5	1	I13	Buccal	Root	Saucer shaped	1 mm
6	1	I14	Buccal	Crown and root	Saucer shaped	<1 mm
7	1	I15	Buccal	Root	Wedge shaped	2 mm
8	1	I17	Buccal	Crown and root	Saucer shaped	<1 mm
9	1	I23	Oral	Crown and root	Wedge shaped	<1 mm
10	1	I25	Buccal	Root	Saucer shaped	<1 mm
11	1	PM7	Buccal	Crown	Wedge shaped	<1 mm
12	1	PM8	Buccal	Crown and root	Wedge shaped	<1 mm
13	2	PM9	Buccal	Crown	Wedge shaped	<1 mm
			Oral	Crown	Wedge shaped	<1 mm
14	1	PM10	Oral	Root	Wedge shaped	<1 mm
15	1	PM11	Buccal	Crown and root	Wedge shaped	<1 mm
16	1	PM13	Buccal	Root	Wedge shaped	>2 mm
17		PM15	Buccal	Root	Saucer shaped	<1 mm
	3	PM15	Oral	Root	Saucer shaped	<1 mm
		PM15	Distal	Root	Saucer shaped	<1 mm
18	1	PM16	Oral	Root	Wedge shaped	<1 mm
19	1	PM17	Buccal	Crown and root	Wedge shaped	<1 mm
20	2	PM19	Buccal	Root	Wedge shaped	<1 mm
		PM19	Oral	Root	Wedge shaped	<1 mm
21	2	PM20	Buccal	Root	Saucer shaped	<1 mm
		PM20	Mesial	Root	Saucer shaped	<1 mm

**Table 2 jpm-13-00772-t002:** NCCL aspects recorded through OCT examination for the lesions not visible at the macroscopic examination.

No.	No. NCCL/Tooth	Tooth with NCCL	Surface with NCCL	Anatomical Location of NCCL	NCCL Shape	NCCL Depth
1	1	I18	Mesial	Root	Saucer shaped	<1 mm
2	1	I24	Distal	Root	Saucer shaped	<1 mm
3	1	I27	Buccal	Root	Irregular	1–2 mm
4	2	PM6	Buccal	Crown and root	Wedge shaped	<1 mm
		PM6	Mesial	Root	Wedge shaped	<1 mm
5	1	PM14	Distal	Root	Saucer shaped	1–2 mm
6	1	PM24	Buccal	Root	Saucer shaped	<1 mm

## Data Availability

The authors declare that the data of this research is available from the corresponding authors upon reasonable request.

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
