# Peer review of "Morphological and Optical Coherence Tomography Aspects of Non-Carious Cervical Lesions"

_jpm, 2023, doi:10.3390/jpm13050772_

Round 1

Reviewer 1 Report

- The findings could be better discussed in relation to clinical applications;

The conclusion has to be more objective and not repeating results. I suggest pointing out the significance of these results in clinical practice.

Author Response

Response to Reviewer 1 Comments

Thank you for the evaluation and for the recommendations!

Point 1: The findings could be better discussed in relation to clinical applications.

Response 1: We have discussed more the results showed in the last figures, pointing out the importance of our findings in clinical practice.

Point 2: The conclusion has to be more objective and not repeating results. I suggest pointing out the significance of these results in clinical practice.

Response 2: According to the recommendations, we have improved the conclusions.

Reviewer 2 Report

The title of the manuscript is appropriate and indicative of the material, which is contained in the manuscript. The abstract is formulated upon all the criteria of the journal and describes clearly the purpose, materials, methods, results and the conclusions. The introduction is clearly formulated, complete and very detailed. There is a clear statement of the objectives of the study. Material and Methods used in the study scientifically valid and technically are correct. The procedures are clearly presented. There is no plagiarism in the article. The statistical analysis is complete, the author describes in a correct way the found result. The results and data gathered in the study are presented in a clear and logical method. The discussion is concisely stated. It develops arguments and theories from evidence. The conclusion is clear, concisely presented and based on the results of the study and on the statistical analysis. In my opinion the article should be accepted.

Author Response

Response to Reviewer 2 Comments

Point 1: The title of the manuscript is appropriate and indicative of the material, which is contained in the manuscript. The abstract is formulated upon all the criteria of the journal and describes clearly the purpose, materials, methods, results and the conclusions. The introduction is clearly formulated, complete and very detailed. There is a clear statement of the objectives of the study. Material and Methods used in the study scientifically valid and technically are correct. The procedures are clearly presented. There is no plagiarism in the article. The statistical analysis is complete, the author describes in a correct way the found result. The results and data gathered in the study are presented in a clear and logical method. The discussion is concisely stated. It develops arguments and theories from evidence. The conclusion is clear, concisely presented and based on the results of the study and on the statistical analysis. In my opinion the article should be accepted.

Response 1: Thank you for evaluating the article and for the appreciations made!

Reviewer 3 Report

The paper is highlighting an important aspect of dentistry, the non-carious tooth loss. The following concerns need further clarification.

1. the title is confusing, use full-form OCT. Also for the first time in the abstract, clarify it in the text.

2. the introduction needs further support in the form of text on loop holes in the scientific data published on the topic. what is the need for doing this study? what is the rationale? the hypothesis of the research.

3. methods, talk about the reliability of the operator, and accuracy of molds or other equipment used in the study, and add a description of statistics especially inferential stats.

4. describe the pictures, with arrows or any suitable means, it is not clear what the author wants to focus a reader on.

Author Response

Response to Reviewer 3 Comments

Thanks for evaluating the article!

The paper is highlighting an important aspect of dentistry, the non-carious tooth loss. The following concerns need further clarification.

Point 1: the title is confusing, use full-form OCT. Also for the first time in the abstract, clarify it in the text.

Response 1: We added the full name of Optical Coherence Tomography in the title and clarified it in abstract and text.

Point 2: the introduction needs further support in the form of text on loop holes in the scientific data published on the topic. what is the need for doing this study? what is the rationale? the hypothesis of the research.

Response 2: The introduction has been developed and it includes now data regarding the terminology of NCCL, the lack of studies regarding the distribution of NCCL on all axial surfaces and the importance of OCT examination in the early diagnostic of dental lesions. We have improved the aim of the study and we have highlighted the need for doing this research.

Point 3: methods, talk about the reliability of the operator, and accuracy of molds or other equipment used in the study, and add a description of statistics especially inferential stats.

Response 3: Material and method - we completed with information about the reliability of the operators, the criteria for identifying NCCL through OCT.

Point 4: describe the pictures, with arrows or any suitable means, it is not clear what the author wants to focus a reader on.

Response 4: Results - We added arrows to identify lesions on macroscopic and OCT images and letters to identify the dental crowns and roots.

Reviewer 4 Report

Comments for authors

I think that the novelty of this paper is the use of OCT. Therefore, it is necessary to state why the use of OCT was conceived, the usefulness of OCT, and a discussion of the results using OCT. also, the criteria of NCCL on OCT analysis should be indicated.

In Figure 1-6, NCCL lesions should be marked with any allow or symbol because some lesions are difficult to find.

Authors should discuss more the result of distribution of NCCL (Figure 7-10).

Authors should provide the type (incisor, canine, premolar, or molar; upper or lower) of all 112 extracted teeth, not just the teeth that had NCCL.

Author Response

Response to Reviewer 4 Comments

Thanks for evaluating the article!

Point 1: introduction must be improved; cited references must be improved; research design must be improved; methods described- can be improved; conclusions must be improved.

Response 1: The introduction has been developed and it includes now data regarding the terminology of NCCL, the lack of studies regarding the distribution of NCCL on all axial surfaces and the importance of OCT examination in the early diagnostic of dental lesions. We have improved the aim of the study and we have highlighted the need for doing this research. Material and method - we completed with information about the reliability of the operators, the criteria for identifying NCCL through OCT. We have discussed more the results showed in the last figures, pointing out the importance of our findings in clinical practice.

Point 2: I think that the novelty of this paper is the use of OCT. Therefore, it is necessary to state why the use of OCT was conceived, the usefulness of OCT, and a discussion of the results using OCT. also, the criteria of NCCL on OCT analysis should be indicated.

Response 2:  We have discussed more about the importance of OCT examination in dental practice in Introduction. In Material and Methods we added the criteria for identifying NCCL through OCT. Also, we have improved the discussion of results obtained using OCT.

Point 3: In Figure 1-6, NCCL lesions should be marked with any allow or symbol because some lesions are difficult to find.

Response 3:  We added arrows to identify lesions on macroscopic and OCT images and letters to identify the dental crowns and roots.

Point 4: Authors should discuss more the result of distribution of NCCL (Figure 7-10).

Response 4: We have discussed more the results showed in the last figures, pointing out the importance of our findings in clinical practice.

Point 5: Authors should provide the type (incisor, canine, premolar, or molar; upper or lower) of all 112 extracted teeth, not just the teeth that had NCCL.

Response 5:  In Material and Methods we added a paragraph in which we provided the type of all the 112 extracted teeth.

Round 2

Reviewer 4 Report

I have read the revised version of manuscript. I think that the authors have revised the manuscript to make it better according to the comments made by the Reviewers. 

I mentioned several comments.

Were there no NCCLs on canines or molars in this study? If so, the authors should revise some of their data.  In general, NCCL-like lesions were more common on canines.

Figure 1-6: Authors should state what the arrows indicate in each figure legend.

P12, L296-: “It was observed that ~ initial examination of teeth.” Why could the OCT detect 7 NCCLs that could not be found by microscopy? What functions of OCT help authors detect NCCL?